# Lipids of Platelet-Rich Fibrin Reduce the Inflammatory Response in Mesenchymal Cells and Macrophages

**DOI:** 10.3390/cells12040634

**Published:** 2023-02-16

**Authors:** Zahra Kargarpour, Layla Panahipour, Michael Mildner, Richard J. Miron, Reinhard Gruber

**Affiliations:** 1Department of Oral Biology, University Clinic of Dentistry, Medical University of Vienna, 1090 Vienna, Austria; 2Department of Pulmonology, University Department of Internal Medicine II, Medical University of Vienna, 1090 Vienna, Austria; 3Department of Dermatology, Medical University of Vienna, 1090 Vienna, Austria; 4Department of Periodontology, School of Dental Medicine, University of Bern, 3012 Bern, Switzerland

**Keywords:** platelet-rich fibrin, mesenchymal cells, macrophages, inflammation, lipids

## Abstract

Platelet-rich fibrin (PRF) has a potent anti-inflammatory activity but the components mediating this effect remain unknown. Blood lipids have anti-inflammatory properties. The question arises whether this is also true for the lipid fraction of PRF. To answer this question, lipid fractions of solid and liquid PRF were tested for their potential to lower the inflammatory response of ST2 bone marrow stromal cells and primary bone marrow macrophages exposed to IL1β and TNFα, and LPS, respectively. Cytokine production and the underlying signalling pathway were analysed by RT-PCR, immunoassays, and Western blotting. We report here that lipids from solid and liquid PRF substantially lowered cytokine-induced expression of IL6, CCL2 and CCL5 in ST2 cells. Moreover, the inflammatory response induced by Pam3CSK4, the agonist of Toll-like receptor (TLR) TLR2, was partially reduced by the lipid extracts in ST2 cells. The PRF lipids further reduced the LPS-induced expression of IL1β, IL6 and CCL5 in macrophages at the transcriptional level. This was confirmed by showing the ability of PRF lipids to diminish IL6 at the protein level in ST2 cells and macrophages. Likewise, PRF lipid extracts reduced the phosphorylation of p38 and JNK and moderately decreased the phosphorylation of NFκB-p65 in ST2 cells. These findings suggest that the lipid fraction is at least partially responsible for the anti-inflammatory activity of PRF in vitro.

## 1. Introduction

Platelet-rich fibrin (PRF) consists of the plasmatic fraction of centrifuged blood [1]. When blood is harvested with clot activator tubes, a coagulated solid PRF forms, while when using plastic tubes, PRF remains liquid for a while [2]. Variations in PRF are further generated by the speed and time of centrifugation [3], as well as the angulation of the rotor [4]. An ideal PRF comprises a high number of platelets and leucocytes suspended in the upper plasma layer [3]. Platelets and leucocytes are considered surrogate parameters to explain the clinical outcome of applying PRF [3], e.g., in socket management and ridge preservation [5], in reducing peri-implantitis defects [6], and for the treatment of refractory leg ulcers [7]. Measuring the platelet-derived growth factors in PRF seems logical considering that, for instance, recombinant transforming growth factor (TGF)-β3 is applied following scar revision surgery [8] and recombinant platelet-derived growth factor-BB (PDGF-BB) therapy was approved for diabetic foot ulcers [9] and used in oral regeneration procedures [10].

Platelet-derived growth factors have extensively been studied not only because they reflect the number of platelets, also because they allow preparing a release kinetic and thereby comparing the various preparations of solid PRF including the classical leucocyte L-PRF and the advanced A-PRFs. The list of platelet-derived growth factors is long and comprises the name-giving PDGF-AA, PDGF-AB, PDGF-BB, as well as TGF-β1, vascular endothelial growth factor, epidermal growth factor, and insulin-like growth factor-1 [11]. Apart from the classical growth factors stored in the platelet granules, bone morphogenetic protein-2 with its osteoinductive properties was identified by immunoassay in PRF [12,13]. Likewise, liquid PRF was characterized by immunoassay of growth factors [14]. In addition to the quantitative immunoassays, proteomic analysis revealed the overall spectrum of proteins in PRF. A total of 652 proteins were detected in lysates of PRF membranes [15] and 705 proteins were identified in the secretome of L-PRF membranes [16]; the proteomic analysis, however, only partially confirm findings based on the immunoassay. On the functional level, genomic profiling of gingival fibroblasts triggered by PRF lysates revealed TGF-β1 to be a major driver of gene expression [15] and platelets’ mitogenic and chemotactic activity involves PDGFs and fibroblast growth factor-2 signaling in mesenchymal cells [17,18].

PRF, however, is complex, and its activity is not limited to platelet- and leucocyte-released growth factors. Plasma-derived clotted fibrin and the other extracellular matrix proteins generate an autologous scaffold resembling the blood clot but are devoid of erythrocytes [19,20]. This aspect is relevant, as erythrocytes weaken the biomechanical properties of PRF membranes [21]. Extracellular matrix proteins are more abundant than structural components. Fibrinopeptides A and B being cleaved from fibrinogen during coagulation can harm endothelial cells [22], and fibronectin is required for fibroblast migration onto fibrin clots [23]. Thus, not only platelet- and leucocyte-derived components but also plasma components generating the extracellular matrix of the clotted plasma exert paracrine activity, an activity being released during the formation and degradation of PRF membranes. Moreover, platelets, leucocytes and plasma, and thus also PRF, are sources of bioactive lipids.

Lipids are the main constituent of human plasma apart from proteins, nucleic acids, and carbohydrates. Lipids are notable because of their structural diversity and the sheer number of discrete molecular species [24]. The six main mammalian lipid categories are fatty acyls, glycolipids, glycerophospholipids, sphingolipids, sterol lipids, and prenol lipids [25]. Plasma lipids are solubilized and dispersed through their association with albumin and more complex lipids with plasma lipoproteins [26]. The fibrin clot with platelets is a rich source of lipoproteins [27], and serum, being a major part of PRF, also contains lipids [28]. Fibrin may even support lipid accumulation because it binds lipoprotein A and low-density lipoproteins [29]. These lipoproteins, in turn, may have anti-inflammatory activity.

High-density lipoproteins (HDL) inhibit a subset of LPS-stimulated macrophage genes that regulate the type I interferon response on murine macrophages [30,31]. Various lipid compounds in blood can help prevent atherosclerosis by combating inflammation [32]. For instance, HDL defends the arterial wall by eliminating cholesterol from lipid-rich macrophages [30,31]. Moreover, HDL and low-density lipoproteins (LDL) bind to LPS to prevent it from activating TLR4 [30,33]. This assumption would extend our findings that PRF and serum containing lipids can reduce inflammation in mesenchymal cells and primary macrophages [34,35]. We can thus hypothesize that at least part of the anti-inflammatory activity of PRF is attributed to lipids.

Our group already showed the capacity of PRF and the blood clot to decrease the proinflammatory response of LPS-induced macrophages by reducing IL-1β [36], IL-6 [21,34,36], CCL2 and CCL5 [34] at the transcriptional level and IL-6 at the protein level [21,36]. Additionally, PRF diminished the expression of IL-6 and nitric oxide synthase (iNOS) in bone marrow-derived ST2 cells and 3T3-L1 mesenchymal cells and reduced the phosphorylation of p65 in ST2 cells [34]. Notably, PRF even eliminated hydrogen peroxide-induced toxicity in gingival fibroblasts [37]. Thus, the application of PRF may reduce inflammatory reactions in vitro; however, the effective component in PRF that mediates these functions is not clear. Since there is strong evidence that PRF has anti-inflammatory properties [21,34,36] and it is well documented that blood lipids have a strong anti-inflammatory activity [30,31,33], we hypothesized that the lipid extracts in liquid and solid PRF can attenuate an inflammatory response driven by MAPK [38] and NF-kB [39] signaling in murine stromal cells and primary macrophages.

## 2. Materials and Methods

### 2.1. Preparation of PPP, Buffy Coat, Blood Clot, and PRF Lysates

Volunteers signed informed consent forms, and the ethics committee of the Medical University of Vienna (1644/2018) approved the preparation of PRF. All experiments were performed in accordance with relevant guidelines and regulations and were conducted in accordance with the Declaration of Helsinki (1975), as revised in 2013. To prepare liquid PRF fractions from noncoagulated blood, we collected venous blood from healthy volunteers, three females and three males from 23 to 45 years of age, in plastic tubes (“No Additive”, Greiner Bio-One GmbH, Kremsmünster, Austria) and centrifuged it at 2000× *g* for 8 min (swing-out rotor; Z306 Hermle, Universal Centrifuge, Wehingen, Germany). The uppermost 2 mL of PPP and the 1 mL of buffy coat (BC) were collected and pooled. PRF membranes from coagulated blood were produced using glass tubes with no silica/silicon added (Bio-PRF, Venice, FL, USA) with centrifugation at 1570 RPM for 12 min (RCF-max = 400 g). The PRF clot was separated from the remaining red thrombus and compressed by the compression tray lid. For blood clot preparation, whole blood was left for half an hour to coagulate in the same tubes. Membranes from PRF and blood clot were transferred to serum-free medium (50 mg membrane/1 mL) and subjected to two cycles of freeze–thawing followed by sonication (Sonopuls 2000.2, Bandelin electronic, Berlin, Germany). After centrifugation (Eppendorf, Hamburg, Germany) at 15,000× *g* for 10 min, 1 mL aliquots of lysates were stored at −20 °C for no longer than one month.

### 2.2. Culture and Isolation of Murine ST2 and Bone Marrow-Derived Macrophages

Murine ST2 bone marrow stromal cells were purchased from RIKEN Cell Bank (Tsukuba, Japan). The cells were cultured and expanded in growth Dulbecco’s modified Eagle’s medium (DMEM, Sigma-Aldrich, St. Louis, MO, USA), supplemented with 10% fetal calf serum (Bio&Sell GmbH, Nuremberg, Germany), and 1% antibiotics (Sigma Aldrich, St. Louis, MO, USA) and seeded at 3 × 10^5^ cells/cm^2^ into 24-well plates. Murine primary bone marrow cells were collected from the femora and tibiae of BALB/c mice, 6- to 8-weeks (Animal Research Laboratories, Himberg, Austria). Bone marrow cells were seeded at 4 × 10^6^ cells/cm^2^ into 24-well plates and grown for 5 days in DMEM supplemented with 10% fetal bovine serum, antibiotics and with 20 ng/mL macrophage colony-stimulating factor (M-CSF; ProSpec-Tany TechnoGene Ltd., Ness-Ziona, Israel). This setting was used with and without TNFα and IL1β at 20 ng/mL (both Sigma Aldrich, St. Louis, MO, USA) or Pam3CSK4 at 5 µg/mL (InvivoGen, Toulouse, France) in ST2 cells and LPS from *Escherichia coli* 0111: B41 (Sigma Aldrich, St. Louis, MO, USA) at 100 ng/mL in murine bone marrow cells, to induce an inflammatory response. In indicated experiments the cells were exposed to the lipids at 20% for overnight under standard conditions at 37 °C, 5% CO_2_, and 95% humidity.

### 2.3. Methanol/Chloroform Extraction of Cellular Lipids

Lipids were extracted according to a modified version of Blight and Dyer explained by Gruber et al. [18]. Briefly, 1.0 mL of each fraction was incubated with a 3.75 mL mixture of chloroform/methanol (1:1, Merck, Darmstadt, Germany) for 10 min with continuous shaking. Thereafter, 1.25 mL of chloroform and 1.25 mL of water were added, and after extensive vortexing, the tubes were centrifuged for 10 min at 1400× *g*. The lower phase containing the lipid fraction was evaporated and dissolved in 100 µL of ethanol and stored at −20 °C. For analysis, the fractions were used at a 1:5 ratio. LiPRF, LiPPP, and LiBC are the lipids isolated from solid PRF membranes, liquid PPP and BC, respectively. LiClot reprsents the lipids extracted from coagulated whole blood.

### 2.4. Reverse Transcription Quantitative Real-Time PCR (RT–qPCR) and Immunoassay

For RT–qPCR [40], after overnight stimulation, total RNA was isolated with the ExtractMe total RNA kit (Blirt S.A., Gdańsk, Poland) followed by reverse transcription (LabQ, Labconsulting, Vienna, Austria) and polymerase chain reaction (LabQ, Labconsulting, Vienna, Austria) on a CFX Connect™ Real-Time PCR Detection System (Bio-Rad Laboratories, Hercules, CA, USA). The primer sequences were mIL6-F: GCTACCAAACTGGATATAATCAGGA, mIL6-R: CCAGGTAGCTATGGTACTCCAGAA; mIL1-F: AAGGGTGCTTCCAAACCTTTGAC, mIL1-R: ATACTGCCTGCCTGAAGCTCTTGT mGAPDH-F: AACTTTGGCATTGTGGAAGG, mGAPDH-R: GGATGCAGGGATGATGTTCT; mCCL5-F: CCTGCTGCTTTGCCTACCTC, mCCL5-R: ACACACTTGGCGGTTCCTTC; mCCL2-F: GCTACAAGAGGATCACCAGCAG, and mCCL2-R: GTCTGGACCCATTCCTTCTTGG. The mRNA levels were calculated by normalization to the housekeeping gene GAPDH using the ΔΔCt method. Supernatants were analyzed for IL6 secretion by immunoassays according to the manufacturer’s instructions (R&D Systems, Minneapolis, MN, USA). RT–PCR data are represented in comparison with the untreated negative control, which was considered 1.0 in all the measurements, and therefore there was no need to show it as a separate group. However, in the IL6 ELISA, the absolute amount of secreted protein from the cells was reported, and therefore untreated cells were also considered to show the amount of protein in all the samples and compare the protein concentration.

### 2.5. Western Blot

ST2 cells were seeded at 5 × 10^5^ cells/cm^2^ into 12-well plates. The following day, serum-starved cells were exposed to 20% LiPRF, LiPPP, LiBC and LiClot for 30 min and then exposed to TNFα and IL1β for another 30 min. Extracts containing SDS buffer with protease and phosphatase inhibitors (cOmplete ULTRA Tablets and PhosSTOP; Roche, Mannheim, Germany) were separated by SDS–PAGE and transferred onto PVDF membranes (Roche Diagnostics, Mannheim, Germany). Membranes were blocked, and the binding of the primary antibody phospho-JNK, phospho-p38, phospho-NF-κB p65 and NF-κB p65 (anti-rabbit IgG, 1:1000, Cell Signaling Technology, #4668S, #4511S, #3033, and #8242S respectively) and JNK and p38 (anti-rabbit IgG, 1:1000, Santa Cruz, #sc-474 and #sc-535 respectively) were detected with the secondary antibody labeled with HRP (anti-rabbit IgG, 1:10,000, Cell Signaling Technology, #CS-7074). After exposure to Clarity Wes tern ECL Substrate (Bio-Rad Laboratories, Inc., Hercules, CA, USA), chemiluminescence signals were visualized with a ChemiDoc imaging system (Bio-Rad Laboratories). For densitometric analysis of blots, the WB images were analyzed using Image Lab software (Bio-Rad Laboratories).

### 2.6. Statistical Analysis

All experiments were performed four times. Statistical analysis was performed with the Friedman test for multiple comparisons without correction of *p* values. The results for the treatment groups were compared with those of the positive control group in the experiments. Analyses were performed using Prism v8 (GraphPad Software, La Jolla, CA, USA). Significance was set at *p*  <  0.05.

## 3. Results

### 3.1. Lipids from PRF Fractions Reduce the Expression of Inflammatory Genes in ST2 Cells and Primary Macrophages

We first assessed the effect of lipid extracts from solid PRF, liquid PRF and whole blood clots on the expression of inflammatory-related genes in cultured ST2 cells and primary macrophages. Cells were stimulated with a TNFα and IL1β cocktail or with LPS, respectively, with and without 20% lipid extracts of solid and liquid PRF. Gene expression analysis showed that LiPRF, LiPPP, and LiBC reduced the expression of the IL6, CCL2 and CCL5 genes in ST2 cells (Figure 1A–C). For the LiClot, we observed a tendency towards a reduction in gene expression. Likewise, the lipid fractions, overall, reduced the expression of IL1, IL6 and CCL5 in primary macrophages (Figure 2A–C). To confirm these findings obtained by gene expression analysis, we measured the protein levels of IL6 in the supernatants. In general, the lipid extracts from PRF and the blood clot reduced the production of IL6 in ST2 cells and primary macrophages (Figure 1D and Figure 2D).

### 3.2. Lipids Extracted from PRF Fractions Suppress Pam3CSK-Induced Inflammation in ST2 Cells

To confirm the findings observed for cytokine-induced inflammation, we exposed ST2 cells to Pam3CSK4, an agonist of TLR2, in the presence or absence of 20% LiPRF, LiPPP, LiBC and LiClot. Lipid extracts isolated from all preparations suppressed the inflammatory response provoked by Pam3CSK4 in ST2 cells (Figure 3A–C). Notably, all preparations reduced the IL6 protein levels (Figure 3D). Again, the LiClot only caused a tendency towards are reduction of gene expression. Altogether, these results suggest that the anti-inflammatory effects of lipids extracted from PRF preparations can be extended toward TLR agonists in murine ST2 cells.

### 3.3. Lipid Fractions from PRF Reduced the Phosphorylation of p38, JNK and NF-κB

To further confirm the ability of lipid extracted from solid and liquid PRF fractions to attenuate MAPK and NF-κB p65 signalling, we carried out Western blot analysis. ST2 cells were treated with 20% of LiPRF, LiPPP, LiBC and LiClot, for 30 min, followed by exposure to TNFα and IL1β for another 30 min. LiPRF, LiPPP, LiBC and LiClot could reduce phosphorylation of JNK and p38 and slightly changed phosphorylation of p65 in ST2 cells (Figure 4A). Uncropped Western blot images are added as Appendix A. Signal intensity was measured and quantified compared to the unphosphorylated antibodies (Figure 4B). This finding suggested that lipid extracts isolated from PRF and blood clot attenuate MAPK and at least moderately reduce NF-κB p65 signalling in ST2 cells.

## 4. Discussion

In support of existing knowledge [21,34,35,36], we show that lipid fractions extracted from liquid PRF, solid PRF and blood clot counteract anti-inflammatory responses in murine mesenchymal ST2 cells and primary macrophages of several pro-inflammatory stimuli. The second finding based on phosphorylation of MAP kinases and p65 suggests that the effect of lipid extracts is regulated by MAPK and to some extend NF-κB signalling. The third finding was that the anti-inflammatory activity of the lipid extracts was not restricted to TNFα and IL1β cytokines but extended towards the TLR signalling pathways. This was indicated when ST2 cells were challenged with Pam3CSK4, the agonist of TLR2 and murine bone marrow cells were challenged with LPS, which is the agonist of TLR4. All the lipid extracts push their significant anti-inflammatory activity by reducing the expression of IL6, CCL2 and CCL5 genes.

The anti-inflammatory action of lipid extracts has already been investigated in different biological samples. For example, the lipid fraction of the secretome of γ-irradiated peripheral blood mononuclear cells lowered the inflammatory response of macrophages exposed to the inflammatory cues [41], and inhibited dendritic cell function 32403085, mast cell degranulation and basophil activation [42]. Elsewhere, pre-treatment of benzo pyrene-exposed rats with placental lipid extracts substantially attenuates the levels of pro-inflammatory cytokines such as TNF-α, IL-1β, and IL-6 [43]. Furthermore, serum HDL has been shown to lower the expression of proinflammatory cytokines in LPS-stimulated bone marrow–derived macrophages [31]. Although many studies showed the anti-inflammatory properties of PRF [34,35,36], the main component leading to this effect is not yet well-studied. One assumption could be the plasma lipids can act as an anti-inflammatory source in PRF.

Clinically bringing this knowledge from other inflammatory disorders to the oral environment could be advantageous to give insights into what clinicians know about periodontal disorders and generate possibilities of new approaches for periodontal therapies. Chronic inflammation such as periodontitis or peri-implantitis is still among the major causes of periodontal disease [44]. Application of PRF was effective in reducing bone loss and modulating inflammatory process [45] or reduce inflammation [34,35,36,46]. Future research should determine if the PRF lipids have a clinically relevant concentration and release kinetic to lower local inflammation over an extended time period. The reason why we have used ST2 cell here is that mesenchymal cells are major drivers of the clinical inflammation in periodontitis [47] and because ST2 cells are the reference we have used to show the anti-inflammatory effects of PRF lysates and its fractions [35]. Clinical biopsies also revealed that myeloid cells, including macrophages, are dominant in healthy and diseased oral mucosa—thus primary bone marrow-derived macrophages represent an in vitro model to reflect the clinical situation, an LPS-sensitive model we have established [34,36]. However, we consider implementing peripheral blood mononuclear cells from healthy volunteers in our future research to have a human and not only mouse in vitro read out system.

Further study limitations are that we did not characterize the lipids causing the anti-inflammatory activity on a molecular level. Future lipidomic analyses of PRF will identify lipid candidates with strong anti-inflammatory activities that should be tested in our in vitro models. Examples of this approach are for instance, the secretome of γ-irradiated peripheral blood mononuclear cells exerts anti-inflammatory effects that are mainly attributed to the lipid fraction as shown for mast cells and basophils [42], dendritic cells [48], and macrophages [41] but also here, lipid species responsible for the observed effect remain unclear—even though extensively studied [42]. Another limitation is that we applied only one method of lipid extraction and we did not evaluate the efficiency of this method comparing to other methods available [49]. The concentration of lipid extracted from each fraction was not standardized and the purity of the lipid fractions was not verified. Moreover, lipid extraction is not an aseptic process although the methanol/chloroform solvent does not allow bacterial growth [50]. The levels of plasma lipids, LDLs and HDLs, are highly variable and donor dependent [51]. We did not compare the lipid extracts from different donors which can be considered in further studies. For this pilot study, we used murine in vitro models that need to be confirmed ideally at the preclinical level showing that lipid extracts from PRF can lower the local inflammatory response in vivo. Taken the data obtained together, we can conclude that lipids are among the effective anti-inflammatory components in solid and liquid PRF as well as the whole blood clot.

## Figures and Tables

**Figure 1 cells-12-00634-f001:**
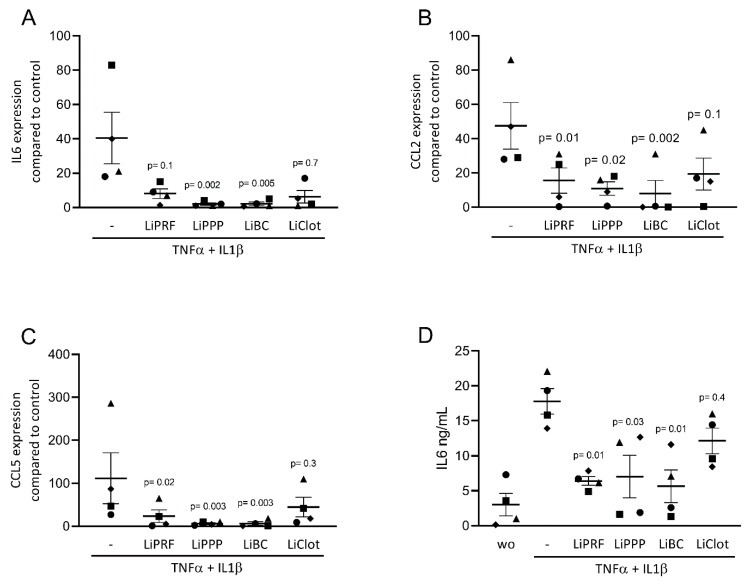
Lipids from PRF reduces the expression of inflammatory-related genes in cytokine-induced ST2 cells. (**A**–**C**) A reduction in IL6, CCL2 and CCL5 in cytokine-induced ST2 cells was reported in the presence of lipid extracts from solid PRF (LiPRF) and liquid PRF (LiPPP and LiBC); with blood clot (LiClot) it was a tendency. (**D**) Data show the IL6 protein levels in the cell supernatant. Each data point represents an independent experiment. N = 4. For comparison of groups, the Friedman test was applied. WO means without and represents unstimulated cells.

**Figure 2 cells-12-00634-f002:**
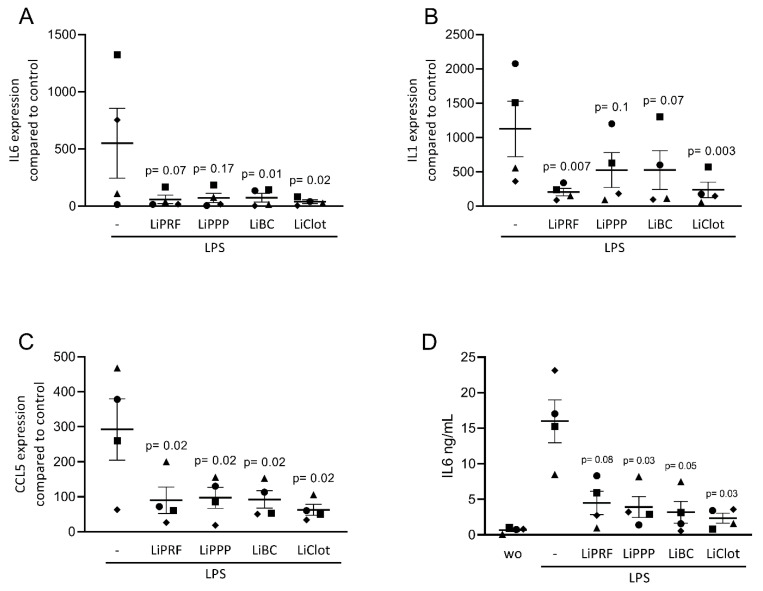
Lipids from PRF reduces the expression of inflammatory-related genes in LPS-treated primary macrophages. (**A**–**C**) Reduction in IL6, IL1, and CCL5 in LPS-treated primary macrophages was reported with lipid extracts from solid PRF (LiPRF), liquid PRF (LiPPP and LiBC) and blood clot (LiClot), mostly reaching the level of significance. (**D**) Data show the IL6 protein levels in the cell supernatant. Each data point represents an independent experiment. N = 4. For comparison of groups, the Friedman test was applied. WO means without and represents unstimulated cells.

**Figure 3 cells-12-00634-f003:**
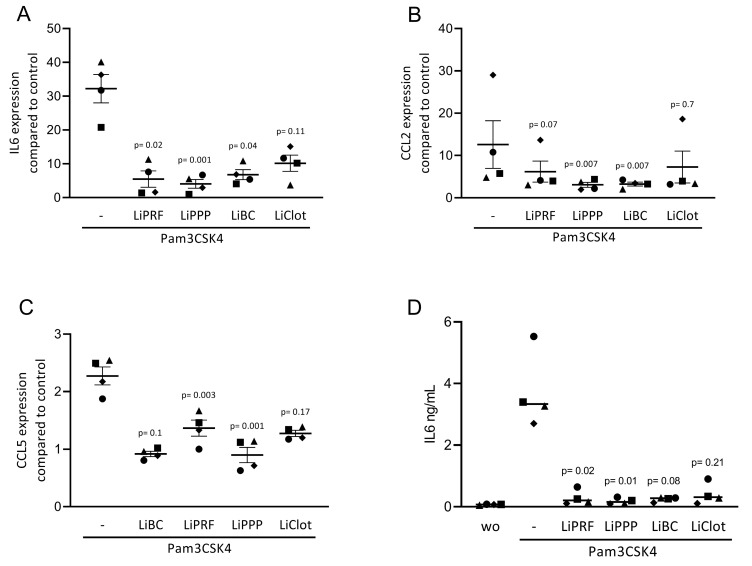
Lipids from PRF can reduce inflammation provoked by TLR2 agonists in ST2 cells. The ST2 cells were exposed to 20% LiPRF, LiPPP, LiBC and LiClot in the presence of 5 μg/mL Pam3CSK4, an agonist of TLR2. (**A**–**C**) Data show the x-fold changes in IL6, CCL2 and CCL5 gene expression and (**D**) the concentration of IL6 in the supernatant of ST2 cells. Each data point represents an independent experiment. N = 4. For comparison of groups, the Friedman test was applied. WO means without and represents unstimulated cells.

**Figure 4 cells-12-00634-f004:**
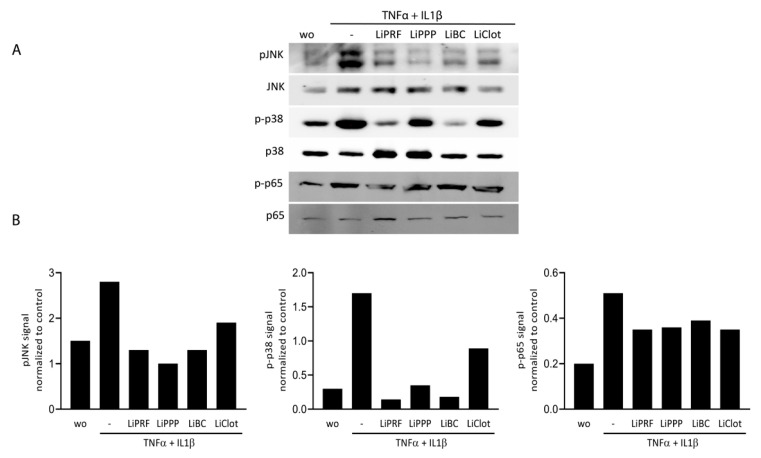
Lipid extracts isolated from PRF and blood reduced phosphorylation of MAP kinases and p65 in ST2 cells. ST2 cells were treated with TNFα and IL1β in the presence or absence of 20% LiPRF, LiPPP, LiBC and LiClot. (**A**) Western blot analysis was carried out for phospho-JNK, -p38, -p65 and total JNK, p38 and p65. (**B**) Western blot signal intensity was quantified for phospho-JNK, -p38, and -p65 compared to total JNK, p38 and p65. WO means without and represents unstimulated cells. Western blot is representative for 2 independent experiments. N = 2.

## Data Availability

The original contributions presented in the study are included in the article/Appendix A. Further inquiries can be directed to the corresponding author.

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
