# Peer review of "Lipids of Platelet-Rich Fibrin Reduce the Inflammatory Response in Mesenchymal Cells and Macrophages"

_cells, 2023, doi:10.3390/cells12040634_

Round 1
Reviewer 1 Report
The current manuscript under review tried to conduct a pilot study to understand if lipid extracts in liquid and solid PRF can attenuate an inflammatory response in murine stromal cells and primary macrophages. Overall, the study is well-designed and reports interesting results aligned with previously published work. The authors are requested to address the following comments before being accepted in the journal.
Provide the details on cell source and its isolation along with the cell culture protocol.
Since the liquid and solid PRFs were obtained from human sources and testing was carried out on murine cells, can the authors speculate on the drawbacks? Please address this limitation of the study while talking about the future direction.
In figure 1d, provide the legend within the figure or in the description for ease of reading. Specifically, it seems like “-“ mean control but what does “wo” mean?
In figures 1 and 2 as well as the results section, it reads “A significant reduction in IL6, IL1, and CCL5 was reported.” However, this is a misleading statement since some of the comparisons resulted in a p-value greater than 0.05. Please rephrase to say that they had a tendency or trend though statistically not significant in certain cases.
Figure 5 requires a scale bar. Provide no primary controls for the immunostaining results in the supplemental section.
Immunofluorescence staining and western blotting results are qualitative in nature and since the goal was to study the unstained vs. stained cells for immunostains and phosphorylation amount in the western blots, please provide a semi-quantitative mode of analysis but providing a ratio. ImageJ can be used for this purpose. Here are some references-
- Ishita Tandon, et al. (2016). Valve interstitial cell shape modulates cell contractility independent of cell phenotype. Journal of
biomechanics, 49(14), 3289–3297. - Iswardy, Edwar, et al. (2017). A bead-based immunofluorescence-assay on a microfluidic dielectrophoresis platform for rapid dengue virus detection. Biosensors and Bioelectronics, 95, 174-180.
- Gallo-Oller, Gabriel, et al. (2018) "A new background subtraction method for Western blot densitometry band quantification through image analysis software." Journal of immunological methods 457, 1-5.
Author Response
Reviewer #1
The current manuscript under review tried to conduct a pilot study to understand if lipid extracts in liquid and solid PRF can attenuate an inflammatory response in murine stromal cells and primary macrophages. Overall, the study is well-designed and reports interesting results aligned with previously published work. The authors are requested to address the following comments before being accepted in the journal.
Provide the details on cell source and its isolation along with the cell culture protocol.
AUTHORS: The changed the cell culture part in methods section as follows: Murine ST2 mesenchymal stromal cells were purchased from RIKEN Cell Bank (Tsukuba, Japan). The cells were cultured and expanded in growth Dulbecco’s modified Eagle’s medium (DMEM, Sigma-Aldrich, St. Louis, MO, USA), supplemented with 10% fetal calf serum (Bio&Sell GmbH, Nuremberg, Germany), and 1% antibiotics (Sigma Al-drich, St. Louis, MO, USA) and seeded at 3 × 105 cells/cm2 into 24-well plates. Murine pri-mary bone marrow cells were collected from the femora and tibiae of BALB/c mice, 6- to 8-weeks (Animal Research Laboratories, Himberg, Austria). Bone marrow cells were seeded at 4 × 106 cells/cm2 into 24-well plates and grown for 5 days in DMEM supple-mented with 10% fetal bovine serum, antibiotics and with 20 ng/mL macrophage colony-stimulating factor (M-CSF; ProSpec-Tany TechnoGene Ltd, Ness‐Ziona, Israel). This setting was used with and without TNFα and IL1β at 20 ng/mL (both Sigma Aldrich, St. Louis, MO, USA) or Pam3CSK4 at 5 µg/mL (InvivoGen, Toulouse, France) in ST2 cells and LPS from Escherichia coli 0111: B41 (Sigma Aldrich, St. Louis, MO, USA) at 100 ng/mL in murine bone marrow cells, to induce an inflammatory response. In indicated experiments the cells were exposed to the lipids at 20% for overnight under standard conditions at 37°C, 5% CO2, and 95% humidity.
Since the liquid and solid PRFs were obtained from human sources and testing was carried out on murine cells, can the authors speculate on the drawbacks? Please address this limitation of the study while talking about the future direction.
AUTHORS: Thanks for the point. We added the sentence for the future studies. Lines 293 to 301: The reason why we have used ST2 cell here is that mesenchymal cells are major drivers of the clinical inflammation in periodontitis [39] and because the ST2 cells are the reference we have used to show the anti-inflammatory erects of PRF lysates and its fractions [28]. Clinical biopsies also revealed that myeloid cells, including the macrophages, are dominant in healthy and diseased oral mucosa – thus primary bone marrow-derived macro-phages represent an in vitro model to reflect the clinical situation, an LPS-sensitive model we have established [27,29]. However, we consider implementing peripheral blood mononuclear cell in our future research to have a human and not only mouse in vitro read out system. Lines 314 and 315: In vitro human models such as PBMCs from the healthy volunteer controls can be also considered in the future studies due to the human origin of the PRF.
In figure 1d, provide the legend within the figure or in the description for ease of reading. Specifically, it seems like “-“ mean control but what does “wo” mean?
AUTHORS: Thanks for the comment. The sentence WO means without and represents unstimulated cells is added to the figure legends for clarity.
In figures 1 and 2 as well as the results section, it reads “A significant reduction in IL6, IL1, and CCL5 was reported.” However, this is a misleading statement since some of the comparisons resulted in a p-value greater than 0.05. Please rephrase to say that they had a tendency or trend though statistically not significant in certain cases.
AUTHORS: Thanks for the comment We changed the reports as indicated.
Figure 5 requires a scale bar. Provide no primary controls for the immunostaining results in the supplemental section.
AUTHORS: Thanks for the comment. The positive controls in figure 5 are not so strong; we thus focus now on WB only by adding MAP kinases phosphorylation and therefore have removed the immunostaining.
Immunofluorescence staining and western blotting results are qualitative in nature and since the goal was to study the unstained vs. stained cells for immunostains and phosphorylation amount in the western blots, please provide a semi-quantitative mode of analysis but providing a ratio. ImageJ can be used for this purpose. Here are some references-
AUTHORS: Thank you for the feedback; we have removed the immunostaining. We used imageLab software to quantify the WB data. We added the WB analysis of pJNK/JNK, p-p38/p38 and new p-p65/p65. The data are now more consistent and it is the p38 and JNK stainings that are convincing.
Ishita Tandon, et al. (2016). Valve interstitial cell shape modulates cell contractility independent of cell phenotype. Journal of
biomechanics, 49(14), 3289–3297.
Iswardy, Edwar, et al. (2017). A bead-based immunofluorescence-assay on a microfluidic dielectrophoresis platform for rapid dengue virus detection. Biosensors and Bioelectronics, 95, 174-180.
Gallo-Oller, Gabriel, et al. (2018) "A new background subtraction method for Western blot densitometry band quantification through image analysis software." Journal of immunological methods 457, 1-5.

Reviewer 2 Report
The manuscript authored by Kargarpour and Gruber investigated the lipid fraction of PRF and its impacts on reducing inflammation. This study further warrants the clinical usage of PRF because not only PRF provides growth factors but also provides anti-inflammatory responses. However there are some issues with justifications and results presentations that will require additional work.
Minor
Page 2 line 71: at least part of the anti-inflammatory activity of PRF is attributed to lipids, not caused by lipids.
Page 2 line 91: are these healthy volunteer smoker s or not? The final lipids fraction comes from pooled PRF or not. These need to be clarified.
Page 3 line 119: what do you mean by saying 20% of the lipids? This is quite confusing in terms of experimental designs.
Page 3 line 144: the untreated control means cells stimulated with cytokines/LPS or completed non-stimulated control? This needs further clarifications.
Major
1) Why CCL2 and CCL5 protein levels were not detected?
2) What are the rationales of using ST2 stem cells and bone marrow derived macrophages from mice, while it is quite convenient to acquire at least PBMCs from the healthy volunteer controls while acquiring your PRF? ST2 stem cell is an interesting selection as stem cells are not the main target for immune regulation? The authors need to expand justifications for using these two cells, otherwise the basic ground of this manuscript is not solid.
3) Another issue is the P-65 detection.
-Quantifications in WB and immunostainings need to be revisited. Particularly for the immunostaining, nuclear positive of p65 need to be analyzed to see the changes.
-if p65 has no changes to reflect the anti-inflammatory function of lipids, how about the MAPK pathway as an alternative? At least some explanations need to be conducted to answer the question of how lipids downregulate the inflammatory cytokines such as IL-6.
Author Response
Reviewer #2
The manuscript authored by Kargarpour and Gruber investigated the lipid fraction of PRF and its impacts on reducing inflammation. This study further warrants the clinical usage of PRF because not only PRF provides growth factors but also provides anti-inflammatory responses. However, there are some issues with justifications and results presentations that will require additional work.
Page 2 line 71: at least part of the anti-inflammatory activity of PRF is attributed to lipids, not caused by lipids.
AUTHORS: Thanks for the comment. We made the change.
Page 2 line 91: are these healthy volunteer smoker s or not? The final lipids fraction comes from pooled PRF or not. These need to be clarified.
AUTHORS: Smoking was not considered as a parameter so we don’t have information about smoking background. The final lipid fractions are pooled. This is added in the method.
Page 3 line 119: what do you mean by saying 20% of the lipids? This is quite confusing in terms of experimental designs.
AUTHORS: We changed the term to ‘’lipids at 20%’’ for more clarity.
Page 3 line 144: the untreated control means cells stimulated with cytokines/LPS or completed non-stimulated control? This needs further clarifications.
AUTHORS: We mean non-stimulated negative control. The change is done.
Major
1) Why CCL2 and CCL5 protein levels were not detected?
AUTHORS: Due to the involvement of IL6 in immune responses and in inflammation, we selected this inflammatory cytokine among all the other detected inflammatory marker genes. The ELISA was in accordance to our previous work. Sure, we could have done also the ELISA for CCL2 and CCL5 but we felt that IL6 is sufficient to support our overall conclusions.
2) What are the rationales of using ST2 stem cells and bone marrow derived macrophages from mice, while it is quite convenient to acquire at least PBMCs from the healthy volunteer controls while acquiring your PRF? ST2 stem cell is an interesting selection as stem cells are not the main target for immune regulation? The authors need to expand justifications for using these two cells, otherwise the basic ground of this manuscript is not solid.
AUTHORS: Chronic inflammation involves mesenchymal lineage, particular fibroblasts, to become a major source of inflammatory mediators (1). Moreover, in our previous study the anti-inflammatory effect of PRF was indicated in ST2 mesenchymal cells and bone marrow primary macrophages – now focusing on the lipid fraction, we have to return to our established models. But indeed, the PBMCs would have been a vital option to prove the anti-inflammatory activity of PRF with cell of human origin. We will take up this recommendation and will take advantage of these cells in future research. We added this aspect as a suggestion for further studies.
3) Another issue is the P-65 detection.
-Quantifications in WB and immunostainings need to be revisited. Particularly for the immunostaining, nuclear positive of p65 need to be analyzed to see the changes.
AUTHORS: Yes, the positive controls are not so strong; we thus focus now on WB only and have removed the immunostaining.
-if p65 has no changes to reflect the anti-inflammatory function of lipids, how about the MAPK pathway as an alternative? At least some explanations need to be conducted to answer the question of how lipids downregulate the inflammatory cytokines such as IL-6.
AUTHORS: Thanks for the comment. We added the data from pJNK and pP38 and repeated P65. Here we show that almost all the lipid fractions can reduce the phosphorylation signal. We added the quantification of the data using imageLab software.
Reference
- Williams DW, Greenwell-Wild T, Brenchley L, Dutzan N, Overmiller A, Sawaya AP, et al. Human oral mucosa cell atlas reveals a stromal-neutrophil axis regulating tissue immunity. Cell. 2021;184(15):4090-104 e15.

Reviewer 3 Report
The study of Kargarpour et al. studied the anti-inflammatory effects of the lipid fraction from PRF. By using ST2 cells and BMDM, authors reported that these lipid fractions diminished the pro-inflammatory response elicited by TNFa+IL1b, Pam3CSK4, and LPS. They further analyzed the cellular mechanism involved, ruling out the intervention of NFKB as the pathway responsible for the observed effects.The manuscript is well written. However, the study lacks novelty, as several studies exist showing that PRF works well as an anti-inflammatory factor. Furthermore, is not clear to me why the authors decided not to assess other pathways involved. It would also be more informative to assess a full spectrum of anti-inflammatory effects and other possible mechanisms supporting the data.
Thus, in this form, the study is only of low interest to the field.
Author Response
Reviewer #3
The study of Kargarpour et al. studied the anti-inflammatory effects of the lipid fraction from PRF. By using ST2 cells and BMDM, authors reported that these lipid fractions diminished the pro-inflammatory response elicited by TNFα+IL1β, Pam3CSK4, and LPS. They further analyzed the cellular mechanism involved, ruling out the intervention of NFKB as the pathway responsible for the observed effects.
The manuscript is well written. However, the study lacks novelty, as several studies exist showing that PRF works well as an anti-inflammatory factor. Furthermore, is not clear to me why the authors decided not to assess other pathways involved. It would also be more informative to assess a full spectrum of anti-inflammatory effects and other possible mechanisms supporting the data. Thus, in this form, the study is only of low interest to the field.
AUTHORS: Thanks for the comment. There are many studies on the anti-inflammatory effect of PRF but none of them directly pointed that this effect can be due to the presence of lipids in the blood. Most of the studies in the PRF relate the activity of PRF to the growth factors and the cells (2, 3). We think that this can be a pioneer study to open a gate for more detailed questions on the essential effects of lipid content of PRF. We added the data from pJNK and pP38 and repeated pP65. Here we show that almost all the lipid fractions can reduce the phosphorylation signal and the reducing effect is regulated by MAP kinases. We added the quantification of the data using imageLab software. In agreement with Reviewer#2, we included WB analysis on p38 and JNK with overall positive findings showing the PRF lipids suppress these pathways.
References
- Dohan DM, Choukroun J, Diss A, Dohan SL, Dohan AJ, Mouhyi J, et al. Platelet-rich fibrin (PRF): a second-generation platelet concentrate. Part II: platelet-related biologic features. Oral Surg Oral Med Oral Pathol Oral Radiol Endod. 2006;101(3):e45-50.
- Fujioka-Kobayashi M, Katagiri H, Kono M, Schaller B, Zhang Y, Sculean A, et al. Improved growth factor delivery and cellular activity using concentrated platelet-rich fibrin (C-PRF) when compared with traditional injectable (i-PRF) protocols. Clin Oral Investig. 2020;24(12):4373-83.

Round 2
Reviewer 1 Report
The resubmission has addressed all my comments. Some minor concerns and one major comment still remain and they are as follows,
Major Concern- What was the sample size in the cell culture experiments performed for carrying out WB and analysis? If it is an n=1, please redo the experiments by increasing the sample size for statistical significance.
Minor Concerns -
1. In the previous set of original comments, the authors changed the corresponding text but are requested to reflect the same in the figure legends. Here is the comment for reference- “In figures 1 and 2 as well as the results section, it reads “A significant reduction in IL6, IL1, and CCL5 was reported.” However, this is a misleading statement since some of the comparisons resulted in a p-value greater than 0.05. Please rephrase to say that they had a tendency or trend though statistically not significant in certain cases.”
2. Check for typographical errors. For instance, does the authors mean "affect" in this following sentence -show the anti-inflammatory erects of PRF lysates
Author Response
Major Concern- What was the sample size in the cell culture experiments performed for carrying out WB and analysis? If it is an n=1, please redo the experiments by increasing the sample size for statistical significance.
AUTHORS Thanks for the comment. We repeated WB twice, n=2.
Minor Concerns -
- In the previous set of original comments, the authors changed the corresponding text but are requested to reflect the same in the figure legends. Here is the comment for reference- “In figures 1 and 2 as well as the results section, it reads “A significant reduction in IL6, IL1, and CCL5 was reported.” However, this is a misleading statement since some of the comparisons resulted in a p-value greater than 0.05. Please rephrase to say that they had a tendency or trend though statistically not significant in certain cases.”
AUTHORS Thanks. The text was changed accordingly.
- Check for typographical errors. For instance, does the authors mean "affect" in this following sentence -show the anti-inflammatory erects of PRF lysates.
AUTHORS Thanks. We checked the whole manuscript again to avoid all the probable typos.

Reviewer 3 Report
Dear Authors,
I have no further comments.
Author Response
Thank you - best regards, Reinhard